# Is the rise in childhood obesity rates leading to an increase in hospitalizations due to dengue?

Chandima Jeewandara[1☯], Maneshka Vindesh Karunananda[1☯], Suranga Fernando[2☯], Saubhagya Danasekara[1], Gamini Jayakody[2], Segarajasingam Arulkumaran[2], Nayana Yasindu Samaraweera[2], Sarathchandra Kumarawansha[2], Subramaniyam Sivaganesh[2], Priyadarshanie Geethika Amarasinghe[2], Chintha Jayasinghe[2], Dilini Wijesekara[2], Manonath Bandara Marasinghe[2], Udari Mambulage[2], Helanka Wijayatilake[2], Kasun Senevirathne[2], Aththidayage Don Priyantha Bandara[2], Chandana Pushpalal Gallage[2], Nilu Ranmali Colambage[2], Ampe Arachchige Thilak Udayasiri[2], Tharaka Lokumarambage[2], Yasanayakalage Upasena[2], Wickramasinghe Pathiranalage Kasun Paramee Weerasooriya[2], Seroprevalence study group[1¶], Graham S. Ogg[3‡], Gathsaurie Neelika Malavige[1,3‡]*

1 Allergy Immunology and Cell Biology Unit, Department of Immunology and Molecular Medicine, University of Sri Jayewardenepura, Nugegoda, Sri Lanka, 2 Ministry of Health, Colombo, Sri Lanka, 3 MRC Translational Immune Discovery Unit, MRC Weatherall Institute of Molecular Medicine, University of Oxford, Oxford, United Kingdom

☯ These authors contributed equally to this work.
‡ These authors are joint senior authors on this work.
¶ Membership of Seroprevalence study group is provided in the Acknowledgements.
* gathsaurie.malavige@ndm.ox.ac.uk

**Data Availability Statement:** Data is available in the manuscript and the supplementary files.

## Abstract

### Background

Obesity and diabetes are known risk factors for severe dengue. Therefore, we sought to investigate the association of obesity with increased risk of hospitalization, as there is limited information.

### Methods and findings

Children aged 10 to 18 years (n = 4782), were recruited from 9 districts in Sri Lanka using a stratified multi-stage cluster sampling method. Details of previous admissions to hospital due to dengue and anthropometric measurements were recorded and seropositivity rates for dengue were assessed. The body mass index (BMI) centile in children aged 10 to 18, was derived by plotting the values on the WHO BMI-for-age growth charts, to acquire the percentile ranking.

### Results

Although the dengue seropositivity rates were similar in children of the different BMI centiles, 12/66 (18.2%) seropositive children with a BMI centile >97th, had been hospitalized for dengue, compared to 103/1086 (9.48%) of children with a BMI centile of <97th. The logistic regression model suggested that BMI centiles 50th to 85th (OR = 1.06, 95% CI, 1.00 to 1.11,

**Funding:** This study has been supported by the World Health Organization Unity Studies (GNM and CJ), a global sero-epidemiological standardization initiative, with funding to the World Health Organization and the UK Medical Research Council (GSO). The World Health Organization unity trial protocol was adopted in trial design. The funders had no role in data collection and analysis, decision to publish, or preparation of the manuscript.

**Competing interests:** The authors have declared that no competing interests exist.

p = 0.048) and BMI centile of >97th (OR 2.33, 95% CI, 1.47 to 3.67, p = 0.0003) was significantly associated with hospitalization when compared to children in other BMI categories.

## Conclusions

Obesity appears to be associated with an increased risk of hospitalization in dengue, which should be further investigated in longitudinal prospective studies. With the increase in obesity in many countries, it would be important to create awareness regarding obesity and risk of severe disease and hospitalization in dengue.

## Author summary

Although obesity and diabetes are known risk factors for severe dengue, there is limited information on whether they are risk factors for increased hospitalization due to dengue. To investigate this, we studied the association of obesity with hospitalization rates for dengue, in children aged 10 to 18 years (n = 4782), who were recruited from 9 districts in Sri Lanka using a stratified multi-stage cluster sampling method. Details of previous admissions to hospital due to dengue and anthropometric measurements were recorded and seropositivity rates for dengue were assessed. The body mass index centile (BMI) in children aged 10 to 18, was derived by plotting the values on the WHO BMI-for-age growth charts, to acquire the percentile ranking. We found that BMI centiles 50th to 85th and BMI centile of >97th were significantly associated with hospitalization rates when compared to children in other BMI categories, which should be further investigated in longitudinal prospective studies.

## Introduction

Dengue is a climate sensitive infection, which was named as 1 of the top 10 threats to global health by the WHO in 2019 [1]. The incidence of dengue is markedly rising in many endemic countries, due in part to intense circulation of multiple dengue virus (DENV) serotypes, increase in global temperatures and erratic rainfall, rapid urbanization and population expansion [2]. Three hundred and ninety million individuals are thought to be infected with the DENV annually, resulting in 100 million symptomatic dengue infections [3]. Although symptomatic dengue is estimated to occur in 1 in 4 of those who are infected with the virus, many studies have reported a wide variability in the ratio of symptomatic: asymptomatic dengue infections. For instance, from 1 in 1.1 to 2.9 in 2004 to 2007 in Thailand [4], 1 in 6.1 in 1980 to 1981 in Thailand [5], 1 in 6 to 13 in Nicaragua [6] and more recently 1 in 1.5 in Indonesia [7]. These differences could be due to different factors such as differences in the virulence of the virus, intense transmission resulting in an increased number of secondary dengue infections associated with a higher risk of severe disease, the interval between infection with different DENV serotypes and host factors [8]. However, there is limited information if host factors such as the presence of comorbidities increase the likelihood of symptomatic infection leading to hospitalizations.

Sri Lanka has experienced dengue outbreaks for over 3 decades, with the incidence rising over time as seen in many countries [9]. The reported cases in Sri Lanka reflects the number of patients who are clinically diagnosed as having dengue and who are hospitalized [9]. In Sri Lanka, as in many other countries, point-of care diagnostic tests such as the dengue NS1

antigen test, or confirmatory tests such as quantitative real-time PCR is not done in public hospitals, due to the non-availability of such tests [10]. Therefore, those who present to out-patient departments with symptomatic dengue infections, who do not require hospitalization are not included in the reported number of cases. The prevalence of diabetes has markedly increased in Sri Lanka, especially in the Western province, where the prevalence of diabetes as risen from 5.02% in 1990, 16.4% in 2006, 27.6% by 2015 to 29% in 2019 [9, 11]. The prevalence of obesity among children also rose from 6.43% in 2003 to by 9.85% 2013, in Colombo, Sri Lanka [9]. Approximately 50% of dengue infections in Sri Lanka are reported from the Western province, which has seen a marked rise over time [9]. Although there could be multiple factors that led to this rise, such as changes in dengue transmission, evolution of the DENV leading to increased virulence, climate change and an increased proportion of those experiencing a secondary dengue infection [10], many host factors could have played a role. For instance, the presence of comorbidities such as obesity, diabetes and renal disease increases the risk of developing severe dengue [8, 12]. As obesity and diabetes are risk factors for occurrence of severe disease, it is possible that they could also lead to an increase in symptomatic/ apparent infection in those infected with the DENV and lead to increase in hospitalizations.

Although obesity is a known risk factor for severe dengue in hospitalized patients [13, 14], whether obesity is a risk factor for an increase in hospitalizations has not been studied. Given the marked rise in obesity, in order to adopt suitable control strategies for dengue, it would be important to find out if obesity indeed increases the risk of hospitalization. Therefore, we investigated if obesity is associated with an increased risk of hospitalization in a large cohort of Sri Lankan children, in an island-wide dengue sero-surveillance study.

## Methods

### Ethics statement

Ethics approval was obtained from the Ethics review committee of University of Sri Jayewardenepura and administrative clearance was obtained from the Ministry of Health Sri Lanka. Informed written consent was obtained from the parents or guardians and assent was obtained from all children. Approval number COVID 12/21.

### Study participants and sampling technique

We carried out an island-wide dengue serosurvey in 4782 school children between the age of 10 and 18 years, who were attending public or private schools in Sri Lanka, during September 2022 to 31st March 2023 [15]. Briefly, healthy children without any comorbidities were recruited following informed written consent from the parents/guardians and assent was taken from children (Table 1). The study was carried out in 9 districts in Sri Lanka, representative of each of the 9 provinces (Fig 1). A stratified multi-stage cluster sampling method was used to select the schools in each district, with a cluster size of 40 students from each cluster. A probability proportionate to the size (PPS) sampling technique was used to select the sample size from each district, as the population size and urbanicity grade varied in different districts. The schools were classified as based in urban, rural or estate areas (tea plantation areas in central highlands) based on the classification from the latest census for Sri Lanka [16].

Anthropometric measurements were obtained at the time the data was collected and blood samples obtained at the schools of the children. The height was measured by a stadiometer to within 0.5cm and weight was measured using a digital scale, which was calibrated regularly throughout the study. In calculating the body mass index centile (BMI) in children aged 10 to 18, the BMI was plotted on the WHO BMI for age growth charts for boys or girls to acquire

**Table 1. Summary of the demographics of the study population.**

|  | Number of Children |
|---|---|
| **Age** |  |
| 10 | 494 |
| 11 | 558 |
| 12 | 544 |
| 13 | 584 |
| 14 | 577 |
| 15 | 605 |
| 16 | 588 |
| 17 | 178 |
| 18 | 654 |
| Total | 4782 |
| **Gender** |  |
| Male | 2240 |
| Female | 2542 |
| Total | 4782 |
| **Seropositivity** |  |
| Negative | 3500 |
| Equivocal | 130 |
| Positive | 1152 |
| Total | 4782 |
| **Urbanization** |  |
| Urban | 840 |
| Rural | 3772 |
| Estate | 170 |
| Total | 4782 |
| **Hospitalized for Dengue** |  |
| Yes | 115 |
| No | 4600 |
| Total | 4782 |
| **BMI Centile** |  |
| <3rd | 1057 |
| 3rd-15th | 939 |
| 15th-50th | 1298 |
| 50th-85th | 862 |
| 85th-97th | 411 |
| >97th | 215 |
| Total | 4782 |
| **District** |  |
| Trincomalee | 236 |
| Polonnaruwa | 231 |
| Jaffna | 297 |
| Matara | 436 |
| Badulla | 478 |
| Ratnapura | 465 |
| Kandy | 608 |
| Kurunegala | 757 |
| Gampaha | 1274 |
| Total | 4782 |

# Map of the study sites of Nine provinces in Sri Lanka

**Fig 1. A map of Sri Lanka showing the locations of recruitment of children from the 9 districts in Sri Lanka.** Each yellow circle corresponds to a study site. The basemap was obtained from the following sources at ArcGIS: Esri, TomTom, Garmin, FAO, NOAA, USGS, OpenStreetMap contributors, and the GIS User Community. https://basemaps.arcgis.com/arcgis/rest/services/World_Basemap_v2/VectorTileServer.

the percentile ranking, as percentile rankings are the most suitable indicator for growth patterns in children [17].

### Determining past dengue disease severity

The parents/guardians of all children who were enrolled in the study were asked to bring all relevant records and diagnosis cards of past hospital admissions, outpatient treatment and clinic attendance. In Sri Lanka, a diagnosis card is issued for each episode of hospitalization, which includes data such as the diagnosis of the illness, relevant clinical, radiological and laboratory findings during hospitalization that supported the clinical diagnosis. Accordingly, details of previous admissions to hospital due to a clinically diagnosed dengue infection were recorded. Those who were found to be seropositive for dengue, but who were not admitted to hospital were considered as not hospitalized due to dengue.

### Assessment of dengue seropositivity

Dengue seropositivity was determined as previously described using a commercial assay (Pan-Bio Indirect IgG ELISA), which has been widely used for dengue seroprevalence studies [18–20]. PanBio units were calculated according to the manufacturer instructions and accordingly, PanBio units of > 11 were considered positive, 9–11 was considered equivocal and < 9 was considered negative.

### Statistical analysis

GraphPad Prism version 9.5 and Jupyter Notebook (python IDE) was used for statistical analysis and to implement models. As the data were not normally distributed, differences in means were compared using the Mann-Whitney U test (two tailed), and the Kruskal-Walli's test was used to compare the differences of the antibody levels in the different districts, and in urban, rural and estate sectors. The degree of associations between BMI, urbanicity and the risk of hospitalization with dengue, was expressed as the odds ratio (OR). Chi Square test was used to determine the association of seropositivity and BMI Centiles of children. The associations between BMI, urbanicity socio-demographic factors to hospitalization status for dengue, was analyzed with the Binary Logistic Regression Model (S1 Data) and implemented using the Synthetic minority oversampling technique (SMOTE) to recover the imbalanced data. The data was considered to be imbalanced as the number of hospitalized children were far fewer than the children who were not hospitalized. Therefore, in order to develop the best performing model, SMOTE model was implemented. Co-morbidities were not assessed in the model as all children were previously healthy apart from varying BMIs.

## Results

### BMI centile and risk of hospitalization due to dengue infection

Overall 1152/4782 (24.1%) were found to be seropositive for dengue and age-stratified seroprevalence rates of each of the districts, and seropositivity rates based on urbanicity has been previously described for this cohort [15]. The number of children enrolled from each district and their demographics are shown in Table 1.

The number of children who had been hospitalized for dengue out of the total number of children who were seropositive for dengue was 182/1152 (15.8%). The BMI centiles of all children aged 10 to 18 (n = 4782), the dengue seropositivity rates of children of different BMIs and hospitalization rates are shown in Table 2. A large proportion (22.1%) of children in Sri Lanka were underweight with their BMIs <3$^{rd}$ centile for age, according to the WHO BMI for age growth

**Table 2. The BMI centile, dengue seropositivity rates and hospitalization rates of children from 9 districts in Sri Lanka aged 10 to 18 years of age.**

| BMI Centile | BMI Centile of Children Aged 10–18 years N = 4782 (%) | Dengue Seropositivity Rates N = 1152 (%) | Hospitalization Rates for Dengue in Dengue Seropositive Children N (%) |
|---|---|---|---|
| <3rd | 1057 (22.10%) | 253 (21.96%) | 22 (8.70%) |
| 3rd to 15th | 939 (19.64%) | 218 (18.92%) | 25 (11.47%) |
| 15th to 50th | 1298 (27.14%) | 284 (24.65%) | 25 (8.80%) |
| 50th to 85th | 862 (18.03%) | 239 (20.75%) | 22 (9.21%) |
| 85th to 97th | 411 (8.59%) | 92 (22.4%) | 9 (9.78%) |
| >97th | 215 (4.50%) | 66 (30.7%) | 12 (18.18%) |

charts for boys or girls [17]. However, 4.5% of children had a BMI of >97th centile for age, when plotted on the WHO BMI for age charts (S1 Table). The dengue seropositivity rates were between 18.9% to 24.6% in children of the BMI groups <97% centile, while the seropositivity rates were 30.7% in those who had a BMI centile of >97th, which were not significantly different (p = 0.16). Of the seropositive children with BMI centile >97th, 12/66 (18.2%) were hospitalized, compared to 103/1086 (9.48%), of children with a BMI centile of <97th (Fig 2). Those with a

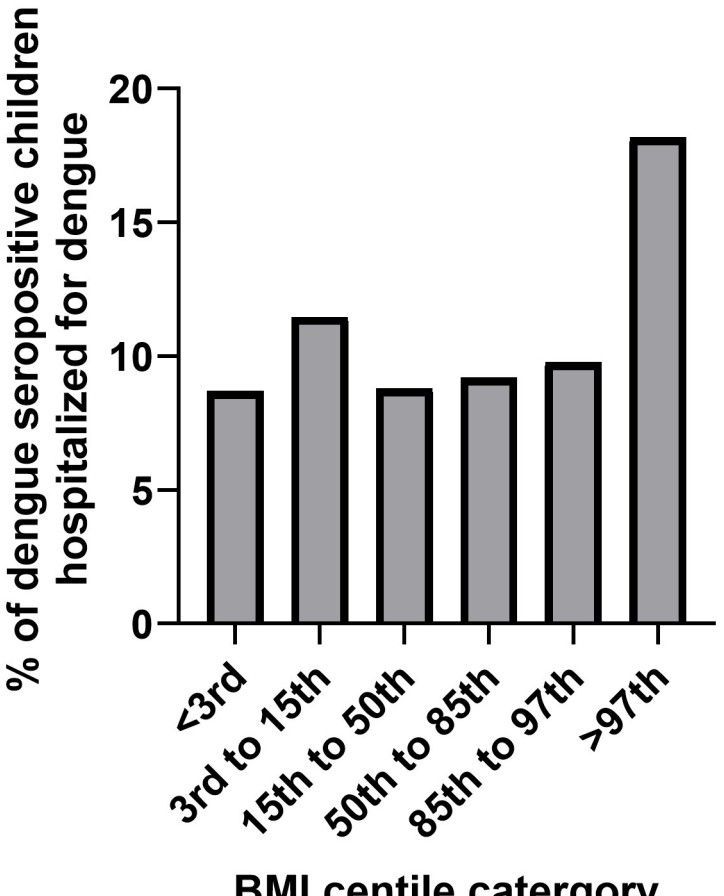

**Fig 2. The proportion of children who were dengue seropositive from admitted to hospital for each BMI category.**

BMI centile of >97th, were twice as likely (odds ratio 2.1, 95% CI, 1.1 to 3.9, p = 0.03) to have been hospitalized for dengue compared to children with a lower BMI. The logistic regression model suggested that BMI centiles 50th to 85th (OR = 1.06, 95% C.I,1.00 to 1.11, p = 0.048) and BMI centile of >97th (odds ratio = 2.33, 95% CI, 1.47 to 3.67, p = 0.0003) was significantly associated with hospitalization when compared to children in other BMI categories.

In addition to the BMI centiles, the risk of hospitalization was significantly higher for females when compared to males (odds ratio = 1.03, 95% CI, 1.00 to 1.06, p = 0.001).

## Urbanicity and risk of hospitalization

Dengue is predominantly an urban infection, as *Aedes aegypti* is the main vector responsible for transmitting dengue along with *Aedes albopictus* [21]. Therefore, we assessed the risk of hospitalization based on the grade of urbanicity (S2 Table). 32/293 (10.9%) dengue seropositive children living in urban areas and 83/859 (9.66%) living in rural and estate areas had been hospitalized for a dengue infection. The implemented logistic model shows that the significant association for urban areas (OR = 1.05, 95% CI, 1.00–1.09, p = 0.015) with the risk of hospitalization with respect to rural areas, although this risk was low.

## Discussion

In this study we have assessed if obesity was associated with an increased risk of hospitalization during an acute dengue infection by assessing hospitalization rates and the BMIs at the time of recruitment to our island wide sero-surveillance study. We found that obese children (BMI centile >97th) were twice as likely to be hospitalized than leaner children. However, although we adopted a novel approach, which uses less resources and time than a prospective, longitudinal observational study, to investigate potential associations between obesity and risk of hospitalization, as this is a retrospective-observational study, there are certain limitations. We only assessed anthropometric measurements at the time of recruitment to this cross-sectional, sero-survey, it would not reflect the BMIs of children at the time of them being infected with the DENV, which is a major limitation of the study. Furthermore, only 12/115 (10.4%) children hospitalized due to dengue had a BMI >97th centile, which suggests that many other factors play a role in symptomatic dengue leading to hospitalization. Nevertheless, although there are many studies that show obesity and diabetes are risk factors for severe dengue in hospitalized patients, if obesity itself leads to increased hospitalization has not been studied. Therefore, despite the limitations, we believe it would be important to explore the findings of our study, by longitudinal study cohorts, to find out if obesity itself was a risk factor for hospitalization and if so the immune mechanisms that lead to this.

Although many high income countries have had the BMIs of their populations rising, the BMIs have plateaued in these countries and in Latin America, while there is a marked and steady rise in the BMI of the population in South Asia and Southeast Asia [22]. The relationship between obesity and development severe dengue had been observed for many years and reported in children in studies in the 1990s from Thailand, India and El Salvodor [23–25]. Many subsequent studies showed that obesity was an independent risk factor for developing severe dengue in hospitalized patients [13, 14] including a recent metanalysis [26]. However, this is the first study reporting that obesity may also associate with higher rates of hospitalization, which needs further examination with longitudinal studies. The number of cases of dengue reported from many countries reflect the number of suspected dengue patients admitted to hospitals, due to the limited availability of point-of care diagnostic tests and confirmatory tests [10, 27]. Therefore, factors that lead to an increase in hospitalizations would also lead to an increase in the reported number of dengue cases in these countries. As there is a marked rise in obesity

in many Asian countries, this could be an additional factor contributing the increase in hospitalization rates, along with intense transmission, co-circulating of multiple DENV serotypes and environmental factors such as climate change, urbanization, and improper waste management.

Obesity is associated with an increase in risk of severe disease due to many other infections such as influenza and COVID-19. While public education programs have focused on the importance of reducing obesity to prevent occurrence of diabetes, cardiovascular diseases and cancer, there has been limited focus on the impact of obesity on many infectious diseases. If the increase in obesity leads to higher hospitalization rates due to dengue, obesity would be an important contributing factor for the current trend in an increase in hospitalizations, experienced in many dengue endemic countries. Therefore, it would be crucial to further investigate the risk of hospitalization due to obesity and to carry out public health campaigns, educating the public on the prevention of obesity. Furthermore, the mechanisms by which obesity and diabetes increase disease severity of dengue, should be further explored to develop biomarkers and therapeutics specially targeting at risk populations.

## Supporting information

**S1 Table. Percentages of children in different BMI centile categories for each of the 9 districts sampled and island-wide.**
(DOCX)

**S2 Table. Dengue seropositivity rates and hospitalisation rates for dengue in dengue seropositive children in urban, rural and estate areas island-wide.**
(DOCX)

**S1 Data. Supplementary data on binary regression model.**
(DOCX)

**S2 Data. Dataset used to generate tables and figures in manuscript.**
(XLSX)

## Acknowledgments

Seroprevalence study group includes the following members:

Lahiru Perera, Pradeep Pushpakumara, Laksiri Gomes, Jeewantha Jayamali, Inoka Sepali Aberathna, Thashmi Nimasha, Madushika Dissanayake, Shyrar Ramu, Deneshan Peranantharajah, Hashini Colambage, Rivindu Wickramanayake, Harshani Chathurangika, Farha Bary, Sathsara Yatiwelle, Michael Harvie, Maheli Deheragoda, Tibutius Jayadas, Shashini Ishara, Dinuka Ariyaratne, Shashika Dayarathna, Ruwanthi Wijekulasuriya, Chathura Ranathunga.

## Author Contributions

**Conceptualization:** Chandima Jeewandara, Gathsaurie Neelika Malavige.

**Data curation:** Chandima Jeewandara, Maneshka Vindesh Karunananda, Suranga Fernando, Saubhagya Danasekara.

**Formal analysis:** Gathsaurie Neelika Malavige.

**Funding acquisition:** Chandima Jeewandara, Graham S. Ogg, Gathsaurie Neelika Malavige.

**Investigation:** Maneshka Vindesh Karunananda, Gamini Jayakody, Segarajasingam Arulkumaran, Nayana Yasindu Samaraweera, Sarathchandra Kumarawansha, Subramaniyam

Sivaganesh, Priyadarshanie Geethika Amarasinghe, Chintha Jayasinghe, Dilini Wijesekara, Manonath Bandara Marasinghe, Udari Mambulage, Helanka Wijayatilake, Kasun Senevirathne, Aththidayage Don Priyantha Bandara, Chandana Pushpalal Gallage, Nilu Ranmali Colambage, Ampe Arachchige Thilak Udayasiri, Tharaka Lokumarambage, Yasanayakalage Upasena, Wickramasinghe Pathiranalage Kasun Paramee Weerasooriya.

**Methodology:** Suranga Fernando.

**Project administration:** Chandima Jeewandara, Maneshka Vindesh Karunananda, Suranga Fernando, Gamini Jayakody, Segarajasingam Arulkumaran, Nayana Yasindu Samaraweera, Sarathchandra Kumarawansha, Subramaniyam Sivaganesh, Priyadarshanie Geethika Amarasinghe, Chintha Jayasinghe, Dilini Wijesekara, Manonath Bandara Marasinghe, Udari Mambulage, Helanka Wijayatilake, Kasun Senevirathne, Aththidayage Don Priyantha Bandara, Chandana Pushpalal Gallage, Nilu Ranmali Colambage, Ampe Arachchige Thilak Udayasiri, Tharaka Lokumarambage, Yasanayakalage Upasena, Wickramasinghe Pathiranalage Kasun Paramee Weerasooriya, Gathsaurie Neelika Malavige.

**Resources:** Chandima Jeewandara, Graham S. Ogg.

**Software:** Saubhagya Danasekara.

**Supervision:** Chandima Jeewandara.

**Writing – original draft:** Gathsaurie Neelika Malavige.

**Writing – review & editing:** Maneshka Vindesh Karunananda, Graham S. Ogg, Gathsaurie Neelika Malavige.

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
