## [Decision Letter · Decision Letter 0]

18 Jan 2024

Dear Professor Malavige,

Thank you very much for submitting your manuscript "Are the rise in childhood obesity rates leading an increase in hospitalizations due to dengue?" for consideration at PLOS Neglected Tropical Diseases. As with all papers reviewed by the journal, your manuscript was reviewed by members of the editorial board and by several independent reviewers. In light of the reviews (below this email), we would like to invite the resubmission of a significantly-revised version that takes into account the reviewers' comments. 

Editor Comments:

I strongly encourage the authors to address each of the reviewer's observations point by point (you can refer to previous responses when appropriate), as there was considerable consistency among the reviews.

A major issue for me was a lack of details on the retrospective analysis. It is important to know how far back (years) were "hospitalized" cases included: 10 years, 5 years, 2 years, 1 year. I do not need to say that your IgG ELISA probably indicates exposure in the past, but is not necessarily associated with the hospitalization record. Was there definitive laboratory confirmation of a dengue case at the time of admission and if so how. Could you distinguish between patients who sought care but were managed as outpatients -- that is your IgG positive children without hospitalization should be called non-hospitalized, they could have had a range of symptoms at home including inapparent infections. Again the concerns that BMI and data collected during the cross-sectional survey would be consistent with that observed in the past when the child had dengue. Also think the suggestions for a more sophisticated multivariate statistical approach is warranted. I appreciated the creative approach (retrospective approach) your research group took to examine the role of obesity in dengue severity. If retrospective review of hospital records (not just parents records - unless they are complete), would provide make this a lot more convincing.

I look forward to a revised manuscript.

We cannot make any decision about publication until we have seen the revised manuscript and your response to the reviewers' comments. Your revised manuscript is also likely to be sent to reviewers for further evaluation.

Sincerely,

Amy C. Morrison, PhD

Section Editor

Andrea Marzi

Section Editor

Editor Comments:

I strongly encourage the authors to address each of the reviewer's observations point by point (you can refer to previous responses when appropriate), as there was considerable consistency among the reviews.

A major issue for me was a lack of details on the retrospective analysis. It is important to know how far back (years) were "hospitalized" cases included: 10 years, 5 years, 2 years, 1 year. I do not need to say that your IgG ELISA probably indicates exposure in the past, but is not necessarily associated with the hospitalization record. Was there definitive laboratory confirmation of a dengue case at the time of admission and if so how. Could you distinguish between patients who sought care but were managed as outpatients -- that is your IgG positive children without hospitalization should be called non-hospitalized, they could have had a range of symptoms at home including inapparent infections. Again the concerns that BMI and data collected during the cross-sectional survey would be consistent with that observed in the past when the child had dengue. Also think the suggestions for a more sophisticated multivariate statistical approach is warranted. I appreciated the creative approach (retrospective approach) your research group took to examine the role of obesity in dengue severity. If retrospective review of hospital records (not just parents records - unless they are complete), would provide make this a lot more convincing.

I look forward to a revised manuscript.

Reviewer's Responses to Questions

**Key Review Criteria Required for Acceptance?**

**Methods**

-Are the objectives of the study clearly articulated with a clear testable hypothesis stated?

-Is the study design appropriate to address the stated objectives?

-Is the population clearly described and appropriate for the hypothesis being tested?

-Is the sample size sufficient to ensure adequate power to address the hypothesis being tested?

-Were correct statistical analysis used to support conclusions?

-Are there concerns about ethical or regulatory requirements being met?

Reviewer #1: -Are the objectives of the study clearly articulated with a clear testable hypothesis stated?

The study aim is well described in the last sentence of the introduction. However, the justification could be strengthened. What additional information will this study provide beyond the existing studies that show an association between obesity and severe disease in hospitalized patients? While I believe it does add value—for instance, this study utilizes a community-based denominator which might be more representative of the general population than hospitalized patients—there are other unique aspects that should be explicitly stated.

-Is the study design appropriate to address the stated objectives?

I found that the methods section lacked the necessary detail to answer this question. For instance, were hospitalizations from any time in the past considered? How can you ensure that the current BMI centile accurately reflects the BMI at the time of admission? Additionally, there is no mention of adjustments for confounders, the most significant of which could be co-morbidities.

-Is the population clearly described and appropriate for the hypothesis being tested?

The population is well described in the first paragraph of the methods and appears to be an appropriate group for this research question. However, dividing the population into 10-18 years and 19-20 years groups is a bit confusing due to the different approaches to scoring obesity. Is this segmentation necessary? If there's no way to apply the same scoring method across the entire age range, perhaps consider only including those aged <=18. The conclusions would remain consistent, and the sample size is still substantial with n=4,782 (1,152 sero-positive). A map of the study sites would be a beneficial addition.

-Is the sample size sufficient to ensure adequate power to address the hypothesis being tested?

With a sample size of n=5,207, it appears adequate for this analysis.

-Were the correct statistical analyses used to support the conclusions?

I'm concerned about the absence of an adjusted analysis. There wasn't any adjustment for age, co-morbidities, or social class—all of which could act as confounders. Additionally, it's worth noting that obesity and these other factors vary over time, implying they might have differed at the time of admission. If an adjusted analysis isn't feasible, then this should be clearly highlighted as a limitation in the discussion section.

-Are there concerns about ethical or regulatory requirements being met?

No.

Reviewer #2: As obesity and diabetes are risk factors for occurrence of dengue severe disease, due to the increasing prevalence of obesity among children in Sri Lanka, the authors hypothesize that this phenomenon also are occurring in the country. 

Consistently, the study population include children aged 10 to 18 , the exposure is the body mass index centile (BMI) and the main outcome hospital admissions. However, in the introduction the authors suggest the potential association between obesity with the risk of symptomatic infection, a driver of hospitalizations, suggesting that obesity could be a risk factor to suffer a symptomatic dengue disease

Also, as end-point the title of the paragraph “determining past dengue disease severity” could be cause confusion about the research question of the study. It´s important. 

Dengue prognosis study require longitudinal design to obtain outcomes as hypotension (determined by age-specific) tachycardia, signs of circulatory insufficiency, any abnormal neurological sign, etc, etc. In this case a cross-sectional will be an inappropriate study design to address this objective.

This study was well conducted using a stratified multi-stage cluster sampling method to select the schools in each district, that is, a method to guarantee a representative sample of the Sri Lanka population, taken account that urbanicity grade varied in different districts. Hopefully, the large sample size will provide an adequate statistical power to examine the questions of interest. The authors should be including the estimates with 95% confidence interval to demonstrate the study has obtained accuracy results thanks to the sample size selected. All of us know the advantages of a confidence interval (rather than a P value) to evaluate research questions. 

The used an objective measure of dengue IgG serostatus to assess whether participants had ever been infected with DENV precludes recall bias. However, the study was conducted during the COVID pandemic period. When you evaluate clinical record, how to differentiate hospitalized Covid-19 from hospitalized dengue cases? any clinical definition? RT-PCR, NS1, IgM test, other methods were conducting for dengue diagnosis? 

Are Outpatients considered inapparent dengue? 

Could you provide information about people who receive medical care and were not hospitalized?

Reviewer #3: Degree of associations between BMI, urbanicity and the risk of hospitalization, was expressed as the odds ratio (OR), there are other statistical methods that could be used to give it greater statistical power. Or use odd ratio adjusted

**Results**

-Does the analysis presented match the analysis plan?

-Are the results clearly and completely presented?

-Are the figures (Tables, Images) of sufficient quality for clarity?

Reviewer #1: -Does the analysis presented match the analysis plan?

Not entirely. I don't see the following analyses described in the "statistical analysis" section in the results:

"As the data were not normally distributed, differences in means were compared using the Mann-Whitney U test (two-tailed), and the Kruskal-Wallis test was used to compare the differences of the antibody levels in the different districts, and in urban, rural, and estate sectors." OR

"Spearman rank order correlation coefficient was used to evaluate the correlation between age and DENV-specific antibody levels (Panbio155 Units)."

Also, the statistical analysis plan doesn't present methods for the "Urbanicity and risk of hospitalization" section in the results.

-Are the results clearly and completely presented?

The text results are sufficient, but some additional tables and figures would help clarify the findings.

 Table 1: A summary of the demographics of the population (traditional Table 1), presenting: Age, district, gender, seropositivity, urbanization, etc.

 Table 2: The current table 1 could be a new table 2.

 Figure 1: A figure showing how the case hospitalization rate varies with BMI centile categorically. With BMI centile category on the X-axis and case hospitalization rate on the Y-axis.

-Are the figures (Tables, Images) of sufficient quality for clarity?

Yes

Reviewer #2: The information provided by the authors is limited . The characteristics of the baseline of the study population should be informed with more detail. Covariates as sex, age (maybe 10-14, 15-18 , more than 18 years), the informant’s education level, any enrolled household member having had a dengue illness diagnosed by a physician either at a hospital , anothers available . I suggest compared, in bivariate analysis, the prevalence of seropositivity by categories of sociodemographic characteristics.

These characteristics are important to appreciate the similarity between groups by covariate’s if not balanced or if the imbalance is not statistically adjusted, these characteristics can cause confounding and can bias study results. This requires adjusted analysis takes into account this baseline between groups that may influence the outcome. 

I cannot identify an specific analysis plan in the manuscript. May be the prevalence ratios could be estimated from generalized estimating equations (GEE) with the Poisson distribution, using the robust sandwich estimate of the variance to account for intra-family correlations

There are not description about the limitations of analysis. The characteristics of the baseline of the population study should be informed in detail to appreciate the differences between groups by covariate.

Reviewer #3: The number of hospitalized people is not very large, they should be very cautious when interpreting the results.

The authors analyze the urbanization variable and the risk of hospitalization. Although they did not find an association when analyzing this variable, it is not well described what health services care is like in urban and rural areas.

**Conclusions**

-Are the conclusions supported by the data presented?

-Are the limitations of analysis clearly described?

-Do the authors discuss how these data can be helpful to advance our understanding of the topic under study?

-Is public health relevance addressed?

Reviewer #1: -Are the conclusions supported by the data presented?

Yes I think so ingeneral, but there a signifant lack of context and depth in the discussion. 

1. Relationship with Previous Studies:

There a some reflection on current literature. However, what do prior studies highlight about other risk factors for hospitalization? How might these factors be realted with obesity? 

2. In-depth Analysis of Results:

It's noteworthy that, while obesity was associated with hospitalization, the vast majority of individuals aged <=18 who were hospitalized fell below the 97th BMI centile (103 out of 115). This observation deserves further elaboration. What implications does this finding have for your overarching conclusions? Further, it's intriguing that only the >97th BMI centile group showcased a heightened admission rate. Why wasn't there a gradational increase observed across the various BMI groups?

-Are the limitations of analysis clearly described?

No, there is no in depth discussion of the limitations of this paper. 

-Do the authors discuss how these data can be helpful to advance our understanding of the topic under study?

Not significantly

-Is public health relevance addressed?

Somewhat, but more detail could be provided. How big a factor is obesity likely to be compared to other risk factors for admis

---

## [Decision Letter · Decision Letter 1]

1 May 2024

Dear Professor Malavige,

Thank you very much for submitting your manuscript "Is the rise in childhood obesity rates leading to an increase in hospitalizations due to dengue?" for consideration at PLOS Neglected Tropical Diseases. As with all papers reviewed by the journal, your manuscript was reviewed by members of the editorial board and by several independent reviewers. The reviewers appreciated the attention to an important topic. Based on the reviews, we are likely to accept this manuscript for publication, providing that you modify the manuscript according to the review recommendations. 

Editor comments:

All reviewers were very appreciative to all of your efforts toward improving your manuscript and although, I'm never supposed to say so directly, this is very close to going to production, but Reviewer #1 still has a few additional queries and suggestions, that I think if addressed with again improve the manuscript. Most of the suggestions are rapid changes and including the logistic regression output as supplemental information is a reasonable request and becoming the new normal in the era of "open" science (I'm still getting used to this).

Again, I will be on alert when the next version comes back and will make the decision myself (no return to reviewers), which was appropriate during the last revision. Congratulations this is almost over the finish line.

Amy

Sincerely,

Amy C. Morrison, PhD

Section Editor

Andrea Marzi

Section Editor

Editor comments:

All reviewers were very appreciative to all of your efforts toward improving your manuscript and although, I'm never supposed to say so directly, this is very close to going to production, but Reviewer #1 still has a few additional queries and suggestions, that I think if addressed with again improve the manuscript. Most of the suggestions are rapid changes and including the logistic regression output as supplemental information is a reasonable request and becoming the new normal in the era of "open" science (I'm still getting used to this).

Again, I will be on alert when the next version comes back and will make the decision myself (no return to reviewers), which was appropriate during the last revision. Congratulations this is almost over the finish line.

Amy

Reviewer's Responses to Questions

**Key Review Criteria Required for Acceptance?**

**Methods**

-Are the objectives of the study clearly articulated with a clear testable hypothesis stated?

-Is the study design appropriate to address the stated objectives?

-Is the population clearly described and appropriate for the hypothesis being tested?

-Is the sample size sufficient to ensure adequate power to address the hypothesis being tested?

-Were correct statistical analysis used to support conclusions?

-Are there concerns about ethical or regulatory requirements being met?

Reviewer #1: -Are the objectives of the study clearly articulated with a clear testable hypothesis stated? YES

- Is the study design appropriate to address the stated objectives? YES (with limitations)

-Is the population clearly described and appropriate for the hypothesis being tested? YES

-Is the sample size sufficient to ensure adequate power to address the hypothesis being tested? YES

-Were correct statistical analysis used to support conclusions? YES

-Are there concerns about ethical or regulatory requirements being met? NO

The methods are significantly clearer.

1. Line197: Worth making it clear to the reader what "imbalanced data" is being referred to here.

2. The authors no present a logistic regression as requested. The reason I had originally requested this was so that co-morbidities could be included in that model. The eligibility excludes children with co-morbidities. I think this is fine, but I would make it clear in the the description of the model within the methods, that co-morbidities were not assessed in the odel as the children were all previous healthy (aprat from varying BMIs).

Reviewer #2: Yes, the hypothesis has been clearly articulated and is verifiable. Although, the best design to answer this research question is a prospective cohort, the authors have declared the limitations of a cross-sectional study to evaluate this research question. The sample was obtained using probabilistic methods, hence the reason why, it is a representative group of the population under study. Additionally, this sampling ensures an adequate sample size for the evaluation of the hypothesis. Finally, the revised manuscript includes a proper statistical analysis to support the discussion.

Reviewer #3: (No Response)

**Results**

-Does the analysis presented match the analysis plan?

-Are the results clearly and completely presented?

-Are the figures (Tables, Images) of sufficient quality for clarity?

Reviewer #1: -Does the analysis presented match the analysis plan? YES

-Are the results clearly and completely presented? NO (see below)

-Are the figures (Tables, Images) of sufficient quality for clarity? YES

Results are also much improved. 

2. The authors should present logistic regression summary tables, ideally in the main mansucript but could be in supplement.

3. I am unclear whether the results of "Urbanicity and risk of hospitalization" are derived from the same logistic regression analysis or a separate one. Could this be made clearer? The model summary tables will help resolve this. The methods mention 4. Spearman rank correlation coefficient, but where are these presented in the results? I mentioned this previously, but perhaps I wasn't clear.

5. Line 198: this is not a predicitive model. I would put a fullstop after model and lose the rest of the sentence.

Reviewer #2: Yes

Reviewer #3: (No Response)

**Conclusions**

-Are the conclusions supported by the data presented?

-Are the limitations of analysis clearly described?

-Do the authors discuss how these data can be helpful to advance our understanding of the topic under study?

-Is public health relevance addressed?

Reviewer #1: -Are the conclusions supported by the data presented? YES

-Are the limitations of analysis clearly described? NO (see below)

-Do the authors discuss how these data can be helpful to advance our understanding of the topic under study? YES

-Is public health relevance addressed? YES

The conclusions are significantly broader and detailed which has strengthened the paper. 

1. Line 258: The authors mention a single weakness, 'However, as we only assessed anthropometric measurements at the time of recruitment'. There are several other weakness, for example this is a retrospecrive and observational study. I would like to see a slightly deeper look at these. Having said this, one of the great strengths of this study, to me, are the innovative (albeit limited) methods. This apporoach does not incur the same time and resource costs as a prospective study and therefore I think this positive point is also woth mentioning. 

2. Finally, I think it is worth noting that only 12/115 hospitalised people were actually in the >97th BMI contile group. I think this statement could be fairly easily integrated into the discussion. I personally think it helps contextualise the findings and reminds us of the importance of non-obesity related factors associated with hospitalisation. The exclusion of individuals with co-morbidities limits the ability of this study to compare the realtive importance of obesity and other co-morbid factors.

Reviewer #2: Yes, this is a relevant public health study. 

It could be inferred that , in children and adolescent ( between 10-18 years old ) , interventions in the lifestyle, diet, physical activity and reduction of sugar intake would be useful to mitigate the risk of hospitalization by dengue. This kind of hypothesis will require different randomized trials.

Reviewer #3: (No Response)

**Editorial and Data Presentation Modifications?**

Reviewer #1: Minor issues: 

Abstract:

- Line 48: Clearly state the aim in the abstract.

- Line 54: BMI should come directly after index. I don't think you mean BMI = "Body Mass Index Centile".

- Line 58: Missing 'Result' subheading. 

- Line 60: Missing 'regression' after logistic.

- Line 63: I think it should be “significantly associated with hospitalisation” not “hospitalisation rates”. This was a patient level analysis. 

Author summary 

This is text recycled from manuscript and should be more aimed at the general public. 

Introduction

- Line 97: 390 starts the sentence, perhaps this should be written out

- Line 99: (and others): Is use of colons better replaced by the word 'in'. e.g 1 in 4 rather than 1:4.

- Line 131: 'with dengue' is unnecessary.

- Line 132: is a bit muddled

- Line 145: Inconsistent use of digits and text for numbers. Here 'nine' is used. In other places numbers are used e.g. line 147. Good to check the whole manuscript. 

- Line 150: What is an 'estate' area.

Methods

- Line 141-2: should be 'between the ages of 10 *and* 18' not '10 *to* 18'.

- Line 162: ethics statement should be moved to the beginning or end of the methods. 

- Line 187: should be 'and *to* implement models' not 'and implement models'

Results

- Line 210: 'and age-stratified seroprevalence rates of each of the districts, and seropositivity rates based on urbanicity has been previously described for this cohort [15]'. Not sure why this is in the results. 

- Line 215: starts with a number.

- Line 222-3: Table 2 doesn't test the significance of this difference, there are no p-values or confidence intervals presented. 

- Line 227: should be 'logistic regression' not just 'logisitic'

- Line 229: 'was significantly associated with hospitalization rates when compared to children in other BMI categories' It should state was significantly associated with hospitalization not 'hospitalization rates' as this is a patient level analysis. Although, we need the model summary tables to be clear about this. 

Discussion

- Line 255: First sentence needs to be a little clearer. '...when infected with DENV...' I don't think is grammatically correct.

- Line 258: should be '...twice as likely...' not '...twice as more likely...'

- Line 270: '...steady rise in BMIs....' perhaps '...steady rise in the BMI of the population in....' or something like like.

Reviewer #2: The authors have carefully studied the comments and recommendations from the reviewers and they have included new data analysis. Aditionally, they offered answers to our questions including new information in the manuscript.

Reviewer #3: (No Response)

**Summary and General Comments**

Reviewer #1: Well done to the authors for making significant improvements to this manuscript. 

I think there are still some minor but essential issues that should be addressed before acceptance. However, I don't think they should be too challeneging to manage. I hope you find the comments helpful. 

As there is no section specfically dedicated to the introduction, I'd like to make some important comments about the introduction:

Line106: “However, if host factors such as the presence of comorbidities increase the likelihood of symptomatic infection leading to hospitalizations has not been studied.” I don’t think this is true. This paper does do this: https://www.sciencedirect.com/science/article/pii/S1201971221006172#sec0006

While this study does not discuss obesity it does look at host factors associated with hospitalisation, which is the claim made by the authors. I don’t think it is essential that this has never been studied for this paper to be valuable and I would strongly recommend removing concrete statements like this. I would rephrase it to say e.g. “there is limited information on…”

-Line131: “Although obesity is a known risk factor for severe dengue in hospitalized patients with dengue[13, 14], whether obesity associates with an increase in hospitalizations has not been studied. The fact that it has never studied is not a sufficient justification for doing the study. I think you just need an additional sentence here (or at the end of paragraph 1) which states the extra value of this study over studies in hospitalised patients, there are several. This is what I was trying to say in my first review, apologies if it wasn’t clear.

Reviewer #2: As I previously wrote , the hypothesis of this study has been clearly articulated and is verifiable.

Also, the authors have declared the limitations of a cross-sectional study to evaluate this research question, the sample was obtained using probabilistic methods and the revised manuscript includes a proper statistical analysis to support the discussion.

If multiple prospective cohort studies in children and adolescents obtain similar results , it would be possible that interventions (in the lifestyle, diets and others ) would be useful to reduce the risk of hospitalization by dengue; randomized trials will be necessary in order to demonstrate this hyphotesis.

Reviewer #3: (No Response)

PLOS authors have the option to publish the peer review history of their article (what does this mean?). If published, this will include your full peer review and any attached files.

Reviewer #1: No

Reviewer #2: No

Reviewer #3: Yes: Crystyan Siles

Figure Files:

While revising your submission, please upload your figure files to the Preflight Analysis and Conversion Engine (PACE) digital diagnostic tool, https://pacev2.apexcovantage.com. PACE helps ensure that figures meet PLOS requirements. To

---

## [Editor Report · Decision Letter 2]

24 May 2024

Dear Professor Malavige,

We are pleased to inform you that your manuscript 'Is the rise in childhood obesity rates leading to an increase in hospitalizations due to dengue?' has been provisionally accepted for publication in PLOS Neglected Tropical Diseases.

Best regards,

Amy C. Morrison, PhD

Section Editor

Andrea Marzi

Section Editor

Congratulations and we appreciate your careful consideration of reviewer feedback.

---

## [Editor Report · Acceptance letter]

31 May 2024

Dear Professor Malavige,

We are delighted to inform you that your manuscript, "Is the rise in childhood obesity rates leading to an increase in hospitalizations due to dengue?," has been formally accepted for publication in PLOS Neglected Tropical Diseases.

Best regards,

Shaden Kamhawi

co-Editor-in-Chief

Paul Brindley

co-Editor-in-Chief
